Population isolation in the Plains spadefoot toad: causes and conservation implications

http://orcid.org/0000-0002-8265-327X Chunco Amanda J. 1 achunco@elon.edu
Nault Emma 1
Silverman Rebecca F. 2
Midolo Sarah 1
Harper Hanna 1
http://orcid.org/0000-0002-5475-8226 Rice Amber M. 2
1 Department of Environmental Studies, Elon University , Elon, NC , United States
2 Department of Biological Sciences, Lehigh University , Bethlehem, PA , United States
Schuster Richard
Electronic publication date: 2024 Oct 7
Publication date: 2024
Volume: 12
Electronic Location ID: e17968
Received 2023 Jul 12; Accepted 2024 Aug 2
Copyright: © 2024 Chunco et al.
Copyright year: 2024
Copyright holder: Chunco et al.
License: This is an open access article distributed under the terms of the Creative Commons Attribution License, which permits unrestricted use, distribution, reproduction and adaptation in any medium and for any purpose provided that it is properly attributed. For attribution, the original author(s), title, publication source (PeerJ) and either DOI or URL of the article must be cited.
License URL: https://creativecommons.org/licenses/by/4.0/

Keywords: Biogeography, Climate change, Conservation, Population genetics, Range disjunctions, Spea bombifrons, Species distribution modeling

Funding: Elon University College of Arts & Sciences and the Department of Biological Sciences at Lehigh University This work was supported by Elon University via the Hultquist Award to Amanda J. Chunco and a SURE Award to Hanna Harper and Sarah Midolo. Undergraduate research funds supporting Rebecca F. Silverman were provided by the College of Arts & Sciences and the Department of Biological Sciences at Lehigh University. Additional research funds were provided to Amber M. Rice by Lehigh University. The funders had no role in study design, data collection and analysis, decision to publish, or preparation of the manuscript.

==============================
Range disjunctions appear to be common in nature, although they may be caused by various factors. They may simply be an artefact of inadequate sampling. If real, they may be the result of colonization events or habitat change. With natural habitats showing increasing fragmentation because of human activity, understanding the cause of a disjunction can have important conservation implications. We investigate the geographical range of the Plains spadefoot toad, Spea bombifrons, a widely distributed species in the midwestern and southwestern United States, with a putative disjunct population in southern Texas. We combine GIS mapping, species distribution modeling, and population genetic analysis to investigate this putative disjunction. We establish that this southern Texas population is truly geographically disjunct and genetically distinct. Further, using climate projections we show that this unique population is at high risk of local extinction.

Introduction

Range disjunctions occur when two or more independent populations of a species are separated by habitats where that species does not occur. This pattern of observed geographical isolation can occur for one of four reasons. First, a rare, long-distance migration event could result in the formation of a new, isolated, population (Dick et al., 2007). Second, a formerly contiguous range could become fragmented as a result of range contractions (Collevatti, Rabelo & Vieira, 2009; Hernández-Roldán et al., 2011). Third, an apparent disjunction could be the result of an error in surveying; the true distribution may be contiguous, but we have incomplete knowledge of the distribution (i.e., the Wallacean shortfall, (Bini et al., 2006)). Finally, the disjunct population could be a cryptic species that has been incorrectly identified.

Range disjunctions are a particular challenge in conservation for several reasons. First, if the disjunction is not recognized, it can lead to an overestimation of a species geographic range size as the species is assumed to occur in areas between known populations (Pena et al., 2014), or an underestimation of genetic divergence as free gene flow between populations is assumed (Llorens et al., 2015). Second, it is often unclear whether a range disjunction is natural or the result of recent human activity. If the former, separate populations should be considered separate management units, while if the latter, restoring population connectivity would be a high conservation priority (Measey & Tolley, 2011). Third, disjunct populations face unique threats due to anthropogenic climate change. Because disjunct populations are, by definition, isolated, individuals may have limited capacity to disperse to more favorable conditions. Simultaneously, the distance between populations means there is little chance for gene flow and thus lower effective population sizes and more limited ability to respond evolutionarily to climate change compared to more widely distributed, connected populations.

Species distribution modeling (SDM) is one approach that has been used to overcome gaps in our knowledge of the distribution of a species. Species distribution models (SDMs) are statistical models that integrate known locality data with presumably relevant ecological data to create maps of potential species distributions (Bradie & Leung, 2017). SDMs have been used in conservation reserve planning (e.g., Carroll, Dunk & Moilanen, 2010), to identify new populations of rare and poorly understood species (e.g., Williams et al., 2009; Chunco et al., 2013), and to predict how distributions may change under different climate change scenarios (e.g., Kearney, Wintle & Porter, 2010). SDMs also have the potential to identify niche similarity between disjunct populations (Shipley et al., 2013) and to provide valuable data on conservation status of species with range disjunctions (e.g., Pena et al., 2014).

Population genetics can also provide information about species ranges. Geographic patterns of population genetic variation can indicate whether populations are located within areas of ancestral range vs. areas of recent expansion (e.g., Rice & Pfennig, 2008), and can provide estimates of interpopulation gene flow and connectivity (e.g., Borokini, Klingler & Peacock, 2021). The relationship between population genetic differentiation and geographic distance would likely differ when pairwise comparisons include a disjunct population vs. when they do not. Specifically, we would expect greater genetic differentiation per unit of geographical distance for comparisons including a disjunct population than for those that do not. Combining niche modeling with population genetics can provide important insights into the mechanisms underlying species range structure and yield important advances in conservation planning (Scoble & Lowe, 2010; Abadía-Cardoso et al., 2021; Lovrenčić et al., 2022).

Here, we focused on Spea bombifrons, a species of spadefoot toad native to the Great Plains region of North America. Although field guides suggest a continuous range across most of the area from west of the Mississippi to the Rockies, and from Canada to Mexico (e.g., Elliott, Gerhardt & Davidson, 2009; IUCN SSC Amphibian Specialist Group, 2015), a disjunct population in southern Texas (TX) that is isolated from the rest of the species’ range has been proposed (e.g., Stebbins, 1985; Gherghel & Martin, 2020) (Fig. 1). Here, we combine GIS mapping, species distribution models (SDMs), and population genetics to address two main questions: 1) Is the southern TX population geographically and environmentally disjunct and genetically distinct?; and 2) How will climate change affect the future distribution of this species?

Figure 1 Range maps of S. bombifrons under two hypothesized distributions.

Under the first scenario (A) the current lack of distributional records reflects a real range disjunction and under the second (B) the apparent gap reflects a lack of adequate sampling while the range is contiguous. The map of North American political boundaries was downloaded from arcgis.com. All symbology used was created within ArcGIS Desktop.

Materials and Methods

Study system

Spea bombifrons is a fossorial amphibian species with a range that extends throughout much of western North America, from southern Canada into northern Mexico (Bragg, 1965; Stebbins, 1985; Elliott, Gerhardt & Davidson, 2009). Although some range maps show a continuous distribution (e.g., Elliott, Gerhardt & Davidson, 2009; IUCN SSC Amphibian Specialist Group, 2015), other sources identify a completely isolated population in southern Texas that is separated from the rest of the range by over 400 km (Stebbins, 1985; Dodd, 2013) (Fig. 1).

In arid regions, S. bombifrons spend most of the year estivating underground, emerging only during monsoon rains in the summer to feed and reproduce (Bragg, 1965). The larval period is very short and can last as little as 2 weeks (Dodd, 2013). Little is known about movement and dispersal in this species, although dispersal is likely limited by their short active period and reliance on ephemeral ponds for moisture and reproduction. Previous population genetic work has shown that S. bombifrons has the highest genetic diversity in the central portion of its range, with lower diversity at the range margins, although southern Texas was excluded from this study (Rice & Pfennig, 2008). This pattern of decreasing genetic diversity at range margins is relatively common (reviewed by Eckert, Samis & Lougheed, 2008) and can result from recent range expansion from a central ancestral population towards the range edges as seen in the spadefoot toads (Rice & Pfennig, 2008) or from decreased population sizes at range margins (Gaston, 2003).

Geographic distribution

To map the known distribution of this species, we used historic museum collection records. Data were downloaded from VertNet (https://vertnet.org/, downloaded on 16 March 2018) or provided directly by museum curators. Records that included collection locality descriptions but were missing latitude and longitude data were assigned initial latitudes and longitudes using the web tool GeoLocate (http://www.geo-locate.org/web/default.html, Rios & Bart, 2010). All records were then manually verified and the precision of the coordinates was estimated to the nearest kilometer following the best practices guidelines established in Chapman & Wieczorek (2006). All records with an accuracy measure of greater than 10 km were discarded. This distance excluded records that were too vague to be useful in identifying the collection locality (e.g., records listing only a state or other large geographic region) while maintaining enough records with reasonably precise locality data (e.g., the level of a specific town or city) to ensure good geographic coverage across the species’ entire range. Additionally, we downloaded research grade observations from iNaturalist for Spea bombifrons (downloaded though GBIF.org (2024) on 07 June 2024) to further demarcate the observed distribution of the species.

Each locality was then mapped in ArcGIS 10.8.2 (ESRI, 2020). The Central Range of S. bombifrons was demarcated by a convex hull polygon added around all locality records outside of South Texas with a 20 km buffer added to account for the fact that the species likely occurred beyond the recorded localities. The South Texas range was similarly demarcated with a convex hull polygon and a 20 km buffer.

As a comparison group to test for the effects of sampling effort and to establish whether other species showed range disjunctions that could represent a geographic barrier to gene flow, we identified 27 additional frog and toad species that occurred in similar habitats to S. bombifrons in Texas. Locality records from those species were downloaded from VertNet (between March and April of 2018) and overlaid on the spadefoot toad distributional map. Of these species, those that did not have any records within at least 20 km of any S. bombifrons locality (indicating potential differences in habitat use) and those species with less than 10 unique localities (indicating rarity) were discarded. Twenty species remained. To simplify the visualization of these species’ ranges, records that fell more than 150 km outside the IUCN defined range of S. bombifrons were removed. Each of these species was then coded for their occurrence within each of the three regions of our study: 1) the Central Range of S. bombifrons; 2) the South Texas range; and 3) the area of hypothesized disjunction between these two regions.

Species distribution modeling

Locality data

The Spadefoot toad locality data obtained from VertNet (as described above) underwent additional cleaning to meet the more stringent requirements of species distribution modeling. Points collected before 1950 and after 2015 were removed to match the timeframe of the climate data used (see below) as closely as possible. Although ideally only using records that matched climate data exactly would be preferable, the reduction in localities would have left several areas of the known range without samples. This broader temporal period ensured a balance between reflecting the best-known range of the species and climatic accuracy. These data were further restricted to those within 5 km of accuracy to ensure that the locality record matched the environmental data. The remaining points were then run through the spatial rarefy tool in SDMtoolbox v2.6 (Brown, Bennett & French, 2017); this removed duplicate localities and reduced the spatial autocorrelation in the collection localities (Brown, 2014).

Environmental data

Spea bombifrons is found in a wide range of habitats, ranging from colder prairie habitat in south-central Canada to the hotter, drier desert habitat of the southwestern US and northern Mexico. This species survives unfavorable conditions through extensive periods of dormancy (Bragg, 1965). Thus, we focused on environmental variables that would be either relevant when the species was active, or related to pond duration, as this is a limiting factor in larval survival (Pfennig & Simovich, 2002). Seven climatic variables were chosen: annual range in temperature, maximum temperature of the warmest month, mean temperature of the wettest quarter, mean temperature of the warmest quarter, precipitation of the wettest month, precipitation of the wettest quarter, and precipitation of the warmest quarter. As the species is only active and only breeds during the summer following heavy rains (Bragg, 1965), these variables were chosen as most relevant to its long-term persistence. These data were downloaded from WorldClim (version 2.0 which compiles climate data from 1970–2000, Fick & Hijmans (2017) at a resolution of 2.5 arcmin). In addition to climate variables, we included average solar radiation in June and July (the months the species is most active across its range) as solar radiation will affect pond duration. These data were also from WorldClim. Finally, we used bulk density (fine earth) of soil at 5 cm, as this measure is important for the fossorial habits of spadefoot toads and will also affect pond duration. These data were from soilgrids.org (Hengl et al., 2017).

Current models

All models were run in Maxent (ver 3.4.4), using a bias file created in SDMtoolbox from the locality records of the 20 comparison species (see above). A bias file is a way to correct for geographic biases in sampling effort; adjusting for sampling bias improves model performance (Syfert, Smith & Coomes, 2013). As the regularization multiplier can strongly impact model output, we evaluated the model outcome across a range of regularization multipliers (0.25, 0.5, 1.0, 2.0, 3.0) to test values both well above and well below the default regularization multiplier, which is 1.0 (Radosavljevic & Anderson, 2014). Because some variables were highly correlated, we ran an additional model with only uncorrelated environmental variables (r ≤ 0.75). These variables were: elevation, bulk soil density, solar radiation in June and July, mean temperature of the warmest quarter, and precipitation of the wettest month. We ran models using either 1) all known localities, 2) localities only from the Central Range, or 3) localities only from South Texas. This allowed us to determine the degree of overlap for the predicted habitat under each case. Each model was replicated 10 times using k-fold cross validation. Finally, we ran a jackknife test of variable importance to measure the contribution of each individual environmental variable to the model.

To evaluate model performance, we used 1) the Area Under the Receiver Operating Characteristic Curve (AUC) and 2) two different independent test files made up of localities not used in model training (Radosavljevic & Anderson, 2014). The first test file was comprised of 63 localities; fifty-nine of these localities came from samples used in a previous study of S. bombifrons (Rice & Pfennig, 2008) while four localities encompassed the additional population genetic sampling for this study described below. The second test file was comprised of the localities downloaded from iNaturalist that we then spatially rarefied at a distance of 5 km to reduce the clustering of localities. We then measured the percentage of samples from both independent test files that were located in ‘suitable’ habitat as predicted by the model. Suitable habitat was defined by applying the 10-percentile training presence threshold to the final model results; this converted the model output from a continuous measure to a binomial map of areas of ‘suitable’ or ‘unsuitable’ habitat. This threshold was selected because the results provide a more conservative test of the population disjunction hypothesis than other commonly used thresholds (e.g., equal training sensitivity plus specificity).

To evaluate the differences in the environment between the Central Range and South Texas, we used the stats base package to run a principal components analysis of the environmental variables and then visualized the results using the ggplot2 package (Wickham, 2016) both using R (ver. 4.4.1, R Core Team, 2024). We then plotted where each toad locality from both populations fell within the PCA to determine whether there was overlap in the environmental conditions between the two habitats.

Future models

We projected the models onto future climate conditions for the year 2070. The year 2070 was chosen to represent a time frame in the near future, while recognizing that some species distributions are already better predicted by 2050 climate data than current climate data (Chunco et al., 2013). We used the GISS-E2-R future climate scenario with a representative concentration pathway (RCP) of 4.5. This RCP represents a future climate change scenario where emissions peak around 2040 and then stabilize. The IPCC considers this an intermediate climate change scenario assuming modest efforts to reduce greenhouse gas emissions (Calvin et al., 2023). We chose this pathway as lower RCPs that will require more stringent emissions policies seem unlikely; this is therefore a conservative choice for conservation planning. These data were downloaded from WorldClim at the same spatial resolution as the current climate data.

Population genetic analyses

We used a complementary S. bombifrons population genetics dataset to determine whether the population genetic signatures are consistent with the South Texas population being genetically distinct from the Central Range. To do this, we combined previously published cytochrome b (cyt b) sequences from S. bombifrons populations sampled within the Central Range (Rice & Pfennig, 2008) with newly generated cyt b sequences from 13 S. bombifrons sampled within the South Texas range. Because we no longer had access to tissue or DNA from the individuals sampled for Rice & Pfennig (2008), sequencing additional markers would only have been possible in our novel South Texas population, and therefore would not have been informative. While sequence analysis of a single mitochondrial marker is not ideal for the characterization of population genetic structure (Toews & Brelsford, 2012), the available genetic data provide an independent but complementary analysis of the S. bombifrons range, together with our species distribution models. We used our expanded S. bombifrons cyt b sequence dataset to construct a haplotype network, calculate pairwise PhiST values among all populations, and test for signatures of isolation by distance.

To add the South Texas S. bombifrons population to previously sampled Central Range populations, we obtained tissue samples from 13 new S. bombifrons individuals. Toe clip tissue samples were collected from 11 juvenile S. bombifrons in May–June 2013 (Brooks County, TX, USA; collected by Gerry Salmon under permit SPI 1097-912) and from one adult S. bombifrons in September 2013 (Hidalgo County, TX, USA; collected by Seth Patterson under a valid Texas State hunting license with Herp Stamp). Toe clip tissue samples were preserved in 70% ethanol. We also obtained one additional Hidalgo County, Texas S. bombifrons tissue sample from the University of Texas at Arlington’s Amphibian and Reptile Diversity Center.

We grouped the Central Range sequences from Rice & Pfennig (2008) plus the new South Texas sequences into 12 populations, each defined by sampling locations within 200-km diameter circles, defined using ArcGIS 10.8.2 (Table S1). Unlike all the other populations, where adult or juvenile toads were sampled, the sequences from the Southeast Arizona population (Rice & Pfennig, 2008) came from tadpoles. Because the vast majority of tadpoles will not survive to reproduce, including all 191 (Rice & Pfennig, 2008) Southeast Arizona cyt b sequences would likely provide an inaccurate representation of population genetic variation. We therefore subsampled this population as follows. The tadpoles had been collected from 16 temporary pond sites, with a maximum of three different haplotype sequences in any individual pond. We chose four samples from each of the pond sites, making sure that we sampled every haplotype. If a pond contained sequences from only a single haplotype, all four samples that we chose had that haplotype. If a pond had three different haplotypes, we sampled all three and chose the fourth from whichever haplotype had the highest frequency among that pond’s samples. We therefore used 64 of the 191 available cyt b haplotype sequences for the Southeast Arizona population. The number of samples from the other 11 populations ranged from 3 to 51, giving us a total of 202 S. bombifrons cyt b sequences (Table S1). Detailed information about all collection locations across both the Central Range and the South Texas range, along with any museum catalog and GenBank accession numbers, is available in Table S1.

We extracted DNA from the new South Texas tissue samples using Qiagen DNeasy Blood and Tissue Kits (Qiagen, Hilden, Germany), following the manufacturer’s protocol. Next, we amplified and sequenced 663 bp of the mitochondrial cyt b gene. We used a forward primer designed from a S. multiplicata cyt b sequence (SCB1-F: 5′-TCCCAACCCCATCTAACATC-3′) and a reverse primer designed from a Xenopus laevis cyt b sequence (XCB2-R: 5′-GAGGGCTAGATTAGGATGGATA-3′), both previously published (Rice & Pfennig, 2008). For PCR amplification, we followed the protocol from Rice & Pfennig (2008), using an Eppendorf Mastercycler Pro gradient thermal cycler (Eppendorf, Hamburg, Germany). After PCR amplification, we purified the PCR products with ExoSAP-IT (Affymetrix, Santa Clara, CA, USA), and submitted sequencing reactions to GENEWIZ (South Plainfield, NJ, USA) for Sanger sequencing. We aligned the new South Texas S. bombifrons cyt b sequences together with the previously published cyt b sequences (Rice & Pfennig, 2008; Table S1) from the 11 other populations using MEGA v6 (Tamura et al., 2013). The new South Texas S. bombifrons cyt b sequences were submitted to GenBank, and the resulting accession numbers are listed in Table S1.

To determine whether South Texas is genetically distinct from the Central Range populations, we used the pegas package (Paradis, 2010) in R to construct a haplotype network using the Parsimony network method (Templeton, Crandall & Sing, 1992). We then used the haplotypes package (Aktas, 2020) in R to calculate pairwise PhiST values for our twelve populations, including 11 within the Central Range and 1 from South Texas. With the ape R package (Paradis & Schliep, 2019), we performed a principle coordinate analysis (PCoA) using the pairwise PhiST values (Borokini, Klingler & Peacock, 2021) to determine whether the South Texas population showed patterns of genetic differentiation from the 11 Central Range populations that would be consistent with population disjunction.

Finally, to assess whether the South Texas population showed greater pairwise genetic differentiation per unit of pairwise geographic distance than the 11 Central Range populations, which would also be consistent with a range disjunction, we assessed isolation by distance relationships both including and excluding all pairwise comparisons involving South Texas. We calculated pairwise geographic distances between the centers of the 200-km diameter circular sampling areas defining all 12 populations by measuring the straight-line distance between each pair of populations in ArcGIS 10.8.2 (Table 1). We then used the vegan package in R (Oksanen et al., 2022) to run Mantel tests on the pairwise geographic distance and pairwise PhiST distance matrices, using the Pearson correlation method and 9,999 permutations. All R analyses were done using version 4.4.1 (R Core Team, 2024). The cyt b sequence data and R code are provided as Supplemental Files.

Table 1 Geographic distances between each Spea bombifrons sampling location, measured between the approximate centers of each 200-km diameter sampling region.

	Northwest Kansas	Northeast Kansas	Southwest Kansas	South Central Kansas	Northwest Oklahoma	Central Oklahoma	East Central New Mexico	Texas Panhandle	West Texas	Southeast Arizona	South Texas	
East Colorado	190 km	750 km	300 km	460 km	480 km	660 km	470 km	650 km	770 km	930 km	1,450 km	
Northwest Kansas		560 km	220 km	310 km	410 km	540 km	540 km	630 km	800 km	1,080 km	1,430 km	
Northeast Kansas			560 km	360 km	560 km	460 km	920 km	820 km	1,050 km	1,510 km	1,430 km	
Southwest Kansas				200 km	200 km	360 km	380 km	410 km	610 km	970 km	1,190 km	
South Central Kansas					210 km	250 km	560 km	480 km	710 km	1,150 km	1,200 km	
Northwest Oklahoma						200 km	400 km	270 km	510 km	970 km	1,020 km	
Central Oklahoma							580 km	390 km	630 km	1,160 km	990 km	
East Central New Mexico								310 km	320 km	600 km	1,040 km	
Texas Panhandle									250 km	790 km	800 km	
West Texas										580 km	740 km	
Southeast Arizona											1,200 km	
South Texas												

Results

Geographic distribution

After removing imprecise locality records, we were left with 5,583 S. bombifrons records from 1,613 unique localities from museums plus an additional 882 observational records from iNaturalist. The distribution map captures the described range of S. bombifrons, from southern Canada into northern Mexico and clearly shows a marked range disjunction between the Central Range and South Texas (Fig. 2A). In the comparison species group, the 20 selected species resulted in 18,323 records covering the entire range of S. bombifrons and the area of disjunction (Fig. 2B). Ten of the 20 species were found across the entire region. Nine species either fell both in the Central Range and region of disjunction (6/20) or in the South Texas range and region of disjunction (3/20). Only one species was restricted to a single region (the Central Range) (Table S2). With locality records from 19 of the 20 comparison species found between the South Texas and Central Range, we conclude that the absence of S. bombifrons is not due to a lack of sampling effort or a geographic barrier, but rather represents a true range disjunction. Furthermore, the fact that no other species shows a similar pattern of disjunction strongly suggests the range disjunction is not due to a common natural barrier to gene flow across many species; it is unique to S. bombifrons.

Figure 2 Map of (A) the point localities of S. bombifrons (light blue points are from vertnet.org, darker blue points are from inaturalist.org, and lavender stars show the location of the population genetic samples), and (B) point localities of the 20 comparison Anuran species.

Species distribution modeling

After cleaning and spatially rarefying the museum locality data to the more precise time frame (1950–2015) and degree of precision (within 5 km), we were left with 923 unique sampling localities: 18 from South Texas and 905 from the Central Range region.

Current models

To evaluate model performance, we compared both AUC values and the omission rate of the independent samples across all models. The regularization multiplier determines how tightly fit the model is to the known presence localities; lower regularization multiplier values produce more tightly fit models which can lead to underprediction of suitable habitat (e.g., overfitting) (Phillips, 2017). Here, we found lower regularization multipliers resulted in higher AUCs and all models correctly identified the presence of most (at least 54 of the 63 points) of the independent population genetic samples and at least 397 out of the 513 points from iNaturalist that remained after spatial rarefication (Table S3). All models confirm that the core of the range is in the Central Range region, with a much smaller region of habitat suitability in South Texas (Figs. 3A–3C). This disjunction is clear across all regularization multipliers; at the default regularization multiplier (1.0), there is a complete gap between these populations where no moderate or high-quality habitat is found (Fig. 3A). This disjunction is even more obvious at the lowest value of the regularization multiplier (0.25); this was also the only model using all locality points to show any high-quality habitat in the south Texas region (Fig. 3B). Even much higher regularization multipliers (3.0) show only lower quality habitat in the region of disjunction (Fig. 3C). As predicted by the literature (Elith et al., 2011; Kramer-Schadt et al., 2013), the reduced variable model showed no substantive differences from the full model (Fig. S1).

Figure 3 The results of Maxent modeling under current environmental conditions with a regularization multiplier of (A) 1.0, (B) 0.25, and (C) 3.0.

Modeling Central Range (Fig. 4A) and South Texas (Fig. 4B) locality records separately further confirms that there is little overlapping habitat space between these two distinct populations. Using only locality data from the Central Range shows no suitable habitat within the entire South Texas region, and using only locality data from South Texas shows no suitable habitat within the Central Range. Additionally, both models also show only low-quality habitat throughout the region of disjunction. As models trained on localities within one region do not identify suitable habitat within the other, this indicates that the environmental conditions between the two regions are distinct and non-overlapping. This finding is further confirmed by our principal components analysis, which shows the two populations do not overlap in environmental space (Fig. 5).

Figure 4 The results of Maxent modeling when (A) only locality records from the Central Range or (B) only locality records from South Texas are used.

Figure 5 The results of the principal components analysis of the environmental variables for each locality where a Spea bombifrons was collected.

Localities from the Central Range are shown in pink and localities from the South Texas population are shown in blue.

When evaluating the effect of individual variables on the model results, soil was the most important variable when using localities from across the geographic range. Soil had an importance value of 25.4% when including all environmental variables and 32.4% in the reduced variable model (Table 2). The regularization multiplier did not have a noticeable impact on the variable contribution. When only the Central Range localities were included, soil was the second most important variable (at 16.1%) while maximum temperature of the warmest month was the most important (20.6%). In contrast, when only South Texas localities were included, elevation was overwhelmingly the most important variable at 71.7%.

Table 2 Variable importance for each of the models run with a regularization multiplier of 1.0.

Variable	Importance value (%)
total range	Importance value (%)
reduced model	Importance value (%)
Central Range	Importance value (%)
South Texas	
Soil	25.4	32.4	16.1	7.0	
Bio 5–max temp of warmest month	21.8		20.6	0.6	
Bio 8–mean temp wettest quarter	13.1		12.2	6.5	
Bio 10–mean temp of warmest quarter	11.1	38.6	14.7	2.7	
Sr 6–solar radiation June	9.2	12.3	15.4	2.0	
Sr 7–solar radiation July	8.9	7.7	6.6	1.7	
Bio 13–precipitation of wettest month	5.9	6.2	4.8	1.5	
Elevation	2.7	2.8	7.4	71.7	
Bio 18–precipitation of warmest quarter	1.2		1.6	2.8	
Bio 16–precipitation of wettest quarter	0.7		0.7	3.5	

Future models

Models using future climate conditions show a significant range retraction across the entire range when all localities are used in training the model across all regularization multipliers (Figs. 6A–6C). The core region of best habitat shows a northward shift with a reduction in overall area, and moderately suitable habitat also shows an extreme reduction. At the default regularization multiplier (1.0), there is the complete loss of any suitable habitat in the South Texas region (Fig. 6A). For both lower (0.25, Fig. 6B) and higher (3.0, Fig. 6C) regularization multipliers, only very small, isolated patches of moderately suitable habitat remain. These models suggest future extirpation of the entire South Texas population.

Figure 6 The results of Maxent modeling under predicted future environmental conditions with a regularization multiplier of (A) 1.0, (B) 0.25, and (C) 3.0.

Population genetic analyses

The haplotype network (Fig. 7) revealed that the South Texas haplotypes are unique to that population, a result that is consistent with the geographic disjunction shown in the species distribution modeling results. The pairwise PhiST values (Table 3) also suggest that South Texas is genetically distinct from the other subpopulations. The PCoA based on these pairwise PhiST values shows that South Texas is differentiated from 10 of the Central Range populations along Axis 1 (Fig. 8), which had an eigenvalue equal to 0.68. Further, South Texas is differentiated from the remaining Central Range population (Southeast Arizona) along Axis 2 (Fig. 8), which had an eigenvalue equal to 0.31.

Figure 7 Genetic relationships and mutational distance among Spea bombifrons cytochrome b haplotypes, and their geographic sampling locations.

(A) Haplotype network for 66 S. bombifrons cyt b sequences. Circles indicate unique haplotype sequences, with size indicating haplotype frequency and color indicating sampling location (see key at upper right for both size and color). Black dots on network branches indicate the number of mutational steps between each haplotype. (B) Geographic sampling locations and frequency for each cyt b haplotype. Each pie chart is centered over a 200-km diameter sampling area (see text). Haplotype identity is represented by the color scheme on the right, with the Roman numerals corresponding to the haplotype labels in the network (A). The size of each section in the pie charts represents the frequency of each haplotype at that sampling location.

Table 3 Pairwise PhiST (Similar to FST, but specifically for sequence data) above diagonal.

Boldfaced PhiST values have significant p-values (p < 0.05) after 9,999 permutations. Actual PhiST p-values for each pairwise comparison are recorded below the diagonal.

	East Colorado	Northwest Kansas	Northeast Kansas	Southwest Kansas	South Central Kansas	Northwest Oklahoma	Central Oklahoma	East Central New Mexico	Texas Panhandle	West Texas	Southeast Arizona	South Texas	
East Colorado		0.429	0	0.311	0.178	0.453	0.144	0.249	0.265	0.471	0.744	0.859	
Northwest Kansas	0.0424		0	0	0	0.425	0	0.064	0.380	0.167	0.791	0.825	
Northeast Kansas	0.4294	0.9999		0	0	0.247	0	0	0.153	0	0.743	0.804	
Southwest Kansas	0.0040	0.9999	0.9999		0	0.203	0	0.063	0.224	0.091	0.744	0.747	
South Central Kansas	0.0015	0.9125	0.9999	0.9491		0.149	0	0.078	0.164	0.096	0.670	0.672	
Northwest Oklahoma	<0.0001	0.0681	0.1727	0.0699	0.0214		0.079	0.066	0	0.070	0.768	0.799	
Central Oklahoma	0.0076	0.6980	0.9999	0.8175	0.7775	0.1057		0.024	0.092	0.031	0.682	0.685	
East Central New Mexico	0.0070	0.4133	0.8771	0.1850	0.0792	0.1219	0.2642		0.108	0	0.692	0.670	
Texas Panhandle	0	0.0309	0.1422	0.0149	0.0013	0.5989	0.0468	0.0731		0.126	0.717	0.778	
West Texas	0.0084	0.4017	0.9999	0.2365	0.2016	0.2574	0.4338	0.9999	0.1753		0.763	0.753	
Southeast Arizona	<0.0001	<0.0001	<0.0001	<0.0001	<0.0001	<0.0001	<0.0001	<0.0001	<0.0001	0.0001		0.769	
South Texas	<0.0001	0.0020	0.0014	0.0001	<0.0001	<0.0001	<0.0001	<0.0001	<0.0001	0.0019	<0.0001		

Figure 8 Results of the principal coordinate analysis (PCoA) of the pairwise PhiST values for the 12 Spea bombifrons populations.

Populations are labeled with their abbreviation (see Table S1). South Texas (STexas) is differentiated from all other populations, which is consistent with the geographic disjunction predicted by the species distribution models.

Our isolation by distance analyses suggest that South Texas shows greater genetic differentiation per unit of geographical distance than the Central Range populations, which is also consistent with the geographic disjunction shown by the species distribution modeling. Signatures of isolation by distance were significant, both when including (Mantel r = 0.72, p = 0.005) and excluding (Mantel r = 0.58, p = 0.04) pairwise comparisons involving South Texas. The slope of the line illustrating the correlation between geographic distance and genetic distance is steeper when the comparisons involving South Texas are included (Fig. 9). Further, the South Texas comparisons all fall above the line that illustrates the isolation by distance correlation among only the Central Range populations (Fig. 9).

Figure 9 Isolation by distance when including and excluding the pairwise comparisons involving South Texas.

The correlation between geographic distance and genetic distance (pairwise PhiST) is significant both when including (r = 0.72, p = 0.005) and when excluding South Texas comparisons (r = 0.58, p = 0.04). South Texas pairwise comparisons (filled circles) all lie above the dotted line depicting the correlation among only the Central Range populations; this is consistent with South Texas showing greater genetic differentiation per unit of geographical distance than the other 11 populations do on average.

Discussion

Evaluating the probability of a true Spea bombifrons range disjunction

Here, we addressed two key questions: 1) Is the South Texas Spea bombifrons population geographically and environmentally disjunct and genetically distinct from the rest of the range? and 2) How will climate change affect the future distribution of this species? We find that all lines of evidence, including the distribution of known locality data (Fig. 2A), the species distribution modeling results (Fig. 3), and the population genetic analyses (Figs. 7–9, Table 3), strongly suggest that the South Texas population of S. bombifrons is completely isolated from the Central Range of the species, consistent with a true range disjunction. The distribution of other amphibian species from throughout the region demonstrate that the gap between populations has been well sampled, so the lack of S. bombifrons records from this region is not the result of under-sampling (Fig. 2B). Further, current climate models (Fig. 3A) and Principal Components Analyses (Fig. 5) suggest this geographic disjunction is likely linked to the environment. Consistent with these findings, our population genetic analyses indicate that S. bombifrons in South Texas are genetically distinct from populations in the Central Range (Figs. 7–9, Table 3).

Current climate models show high habitat suitability throughout the Great Plains, into southern Canada, and in south Texas. Maxent models suggest a very limited potential for connectivity between the two populations if all localities from both the Central Range and South Texas are used (Fig. 3). This lack of connectivity is even more obvious when populations are modeled independently (Fig. 4). Models using both more and less restrictive regularization multipliers show similar patterns (Figs. 3B and 3C).

Several biologically important variables contributed to these results. Bulk soil density was particularly important as it had the highest percent contribution to the models (Table 2). Although soil variables can improve model performance in woody plant species, soil variables still remain rarely used in species distribution studies (Hageer et al., 2017). Given the important contribution of the soil variable in our results (Table 2), we would reiterate Hageer et al.’s (2017) suggestion that soil variables be more widely adopted in species distribution modeling, even for animal studies. The inclusion of soil variables is likely to be particularly important for fossorial animal species such as Spea bombifrons.

Several independent population genetics analyses are consistent with the disjunction of the South Texas population. The South Texas cytochrome b haplotypes are unique to this population (Fig. 7), and this population is differentiated from the remaining populations (Fig. 8, Table 3). Also, the isolation by distance analyses suggest that for the cyt b locus, South Texas shows greater genetic differentiation per unit of geographical distance than the Central Range populations (Fig. 9), again consistent with population disjunction.

As none of the other frog and toad species we examined show a similar pattern of disjunction (Fig. 2B), the isolation of the South Texas population likely occurred due to a rare long distance colonization event rather than a major habitat change causing an overall range retraction and a residual relict population. However, Gherghel & Martin’s (2020) study of glacial refugia in Spea spp. suggested that there may have been multiple refugia for S. bombifrons. Our work does not allow us to rule out the possibility that ancient habitat change caused a range retraction specific to S. bombifrons.

Predicting climate change impacts on future species distribution

Projecting the future range of S. bombifrons based on climate models shows troubling results. Under even the conservative representative concentration pathway chosen for future greenhouse gas emissions, range retractions are predicted across the entire range of the species. Furthermore, the isolated South Texas population will likely be severely reduced or even extirpated entirely (Fig. 6). While many species are responding to climate change by shifting ranges northwards (Parmesan, 2006; Angert et al., 2011; Chunco, 2014), the limited dispersal ability of this species combined with the unsuitable habitat underlying the range disjunction makes it unlikely that the South Texas population will be able to respond to climate change via dispersal.

As anthropogenic forces contribute to increasing range disjunction in many species as a result of habitat fragmentation and loss, predicting how individual populations will respond to climate change is increasingly important. The South Texas population of spadefoot toads experiences a significantly different climate than other populations and has two unique haplotypes not seen in other populations (Fig. 7). Future studies that attempt to detect signatures of selection across the entire genome for S. bombifrons from South Texas could provide some insight into whether and how local adaptation of the South Texas population may have contributed to its survival in this region.

Understanding the geographic distribution of a species and the cause of any disjunction is critically important in predicting the response of that species to climate change. For example, anthropogenically driven disjunctions due to land use conversion and the resulting habitat fragmentation can greatly limit a species’ ability to respond to climate change by restricting available habitat to shift into and because isolated populations have lower standing genetic variation and thus a decreased ability to adapt to climate change. Significant literature has focused on the impacts of anthropogenic habitat fragmentation on the abilities of species to respond to climate change (e.g., Honnay et al., 2002; Opdam & Wascher, 2004).

Natural (non-anthropogenic) range disjunctions can also be important considerations in evaluating and predicting a species’ response to climate change. For example, climate change will proceed at a different rate for populations occurring at different latitudes or elevations, and population isolation means that any given population will have a lower effective population size than an overall census would suggest. Thus, the combination of limited geographic range, low gene flow between populations, and demographic stochasticity within populations may mean that the extinction risk of disjunct populations is higher than currently recognized.

Conclusions

We used GIS mapping, species distribution modeling, and population genetics to investigate a proposed range disjunction in the Plains spadefoot toad, Spea bombifrons. We first aimed to determine whether a population in southern Texas is geographically and environmentally disjunct and genetically distinct from the rest of the S. bombifrons range. We then explored how climate change is likely to affect the distribution of this species in the future. Our results indicate that the S. bombifrons population in southern Texas is isolated both geographically and genetically from other populations of this species, likely as a result of unsuitable environmental conditions. As such, we conclude this population of Plains spadefoot toads represents a true range disjunction. Further, after projecting our modeling results onto predicted future climate conditions for the year 2070, we predict widespread range retractions for S. bombifrons, with a high likelihood of local extinction for the southern Texas population. Our results underscore the importance of considering range disjunctions and population isolation when planning conservation efforts.

Supplemental Information

Supplemental Information 1 Spea bombifrons population groups for the population genetic dataset, including collection location information, sample size, any museum catalog numbers, and GenBank Accession numbers.

Museum catalog abbreviations: OMNH: Sam Noble Oklahoma Museum of Natural History, University of Oklahoma; FHSM: Sternberg Museum of Natural History, Fort Hays State University; MVZ: Museum of Vertebrate Zoology, University of California, Berkeley; TNHC: Texas Science & Natural History Museum, University of Texas at Austin; UTA A, Amphibian and Reptile Diversity Research Center, University of Texas at Arlington.

Supplemental Information 2 The regions where the comparison species have been observed based on museum data.

Supplemental Information 3 Model validation metrics.

All models were replicated 10 times using cross-validation and the mean results were submitted. The two independent test samples came from the iNaturalist data and haplotype data as described in the main text.

Supplemental Information 4 The results of MaxEnt Modeling using only uncorrelated variables.

All localities and the default regularization multiplier (1.0) were used.

Supplemental Information 5 Data from iNaturalist downloaded from gbif.org.

Supplemental Information 6 The population pairwise PhiST values.

Supplemental Information 7 The population pairwise geographic distance values that are used in the isolation by distance analysis.

Supplemental Information 8 Locality data for the species used to create the bias file and the map of collection localities.

Supplemental Information 9 R code used for constructing the haplotype network and calculating the population genetics statistics for the four S. bombifrons populations.

Supplemental Information 10 Independently collected locality data used to test the maxent model.

Supplemental Information 11 S. bombifrons sequence data.

Supplemental Information 12 Locality data used in Maxent models.

Supplemental Information 13 Locality data of Spea bombifrons.

We thank Gerry Salmon and Seth Patterson for assistance with South Texas S. bombifrons tissue collections, and the University of Texas at Arlington Amphibian and Reptile Diversity Center for granting us a S. bombifrons tissue sample from their collection. We would also like to thank Jennifer Hamel, Jennifer Dabrowski, Kelsey Bitting, Jonathan Su, Christopher Richardson, and Richard Blackmon for feedback on earlier versions of the manuscript. Finally, we would like to thank NimBios for feedback and suggestions that resulted from AJC’s participation in the NimBios: Applications of Spatial Data Workshop and the AMNH for feedback during a workshop on species distribution modeling at the Southwest Research Station.

Additional Information and Declarations

Competing Interests

Author Contributions

DNA Deposition

Data Availability

The authors declare that they have no competing interests.

Amanda J. Chunco conceived and designed the experiments, performed the experiments, analyzed the data, prepared figures and/or tables, authored or reviewed drafts of the article, and approved the final draft.

Emma Nault performed the experiments, analyzed the data, prepared figures and/or tables, and approved the final draft.

Rebecca F. Silverman performed the experiments, analyzed the data, prepared figures and/or tables, and approved the final draft.

Sarah Midolo performed the experiments, analyzed the data, prepared figures and/or tables, and approved the final draft.

Hanna Harper performed the experiments, analyzed the data, prepared figures and/or tables, and approved the final draft.

Amber M. Rice conceived and designed the experiments, performed the experiments, analyzed the data, prepared figures and/or tables, authored or reviewed drafts of the article, and approved the final draft.

The following information was supplied regarding the deposition of DNA sequences:

The newly generated Spea bombifrons cyt b sequences from the South Texas population are available at GenBank: OR256236–OR256248.

The previously published S. bombifrons cyt b sequence data from the Central Range are available at GenBank: EU285613–EU285642, EU499393–EU499394, EU499396–EU499401, EU499404, EU499406, EU499412, EU499415-EU499416, EU499418–EU499427, and EU499429–EU499435.

The sequence data for every sample included here are available in the Supplemental File.

The following information was supplied regarding data availability:

The raw data is available in the Supplemental Files.

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
