# Peer review of "Population isolation in the Plains spadefoot toad: causes and conservation implications"

_PeerJ, doi:10.7717/peerj.17968_

## Round 0.1 · original submission · Major Revisions

The reviewer point out a number of important concerns that need to be addressed. One of the major concerns relates to genetic component, raising doubts if it is adequate in this study considering the limitation in the sampling design (i.e., number of samples used and use of only one uniparental molecular marker). Hence, taken on balance, I strongly recommend authors remove the genetic component of their study. Otherwise, authors must improve the sampling design, which includes adding a set of (uni- and biparental) molecular markers. Please see reviews for more detail and comprehensively address all in your revision and response to the review letter.

· Appeal

Appeal

Thank you very much for considering our manuscript “Population isolation in the Plains spadefoot toad: Causes and conservation implications” for publication in PeerJ (#2023:04:85079:0:0). We received the email regarding the decision to Reject our manuscript September 28. After reading the editor and reviewer comments, we would like to argue for the consideration of a resubmission of our manuscript.

As the reviewer noted, in our manuscript we combined GIS mapping, species distribution modeling, and population genetics to investigate a potential range disjunction in the Plains spadefoot toad. We also perform species distribution modeling under future climate projections to evaluate the extinction risk for a disjunct population of this species located in southern Texas.

The reviewer’s main concerns centered largely on the population genetics data, versus the species distribution modeling. However a review of PeerJ published manuscripts on species distribution modeling shows that our methods in this arena are consistent with literature published in PeerJ; PeerJ publishes many papers that use the species distribution modeling method, and this method alone including by one of the current authors (Archis et al. 2018). In fact, it is rare for such a study to also include population genetic data analyses. One unique aspect of our manuscript is that it does incorporate some population genetics analyses to reinforce the results and conclusions that come out of the species distribution modeling. Therefore, the population genetics component of this manuscript is only a small subset of what is presented, and mainly serves the role of providing independent support for the species distribution modeling results.

We appreciate that the handling editor (Dr. Alison Nazareno) went out of his way to give careful and constructive review, with a focus on his expertise (ecological genetics and genomics). Indeed, according to the comments we received from Dr. Nazareno, his decision to reject our manuscript was focused on the use of a single marker in our population genetics analyses. If our study was purely a population genetics study of the Plains spadefoot toad and its distribution, we would understand Dr. Nazareno’s decision. However, as explained above, the majority of our manuscript is based on the species distribution modeling of this toad species, now and in the future. Our inclusion of any population genetic data goes beyond what is usually done in this field.

We feel that the handling editor(s) should be experts in geospatial modeling, as this is the major thrust of our study. We have heavily edited a revised version to more strongly emphasize the biogeographical focus of our work to allow it to more appropriately be placed in context of the GIS/modeling field, rather than as a population genetics paper.

Additionally, we recognize that a population genetics-focused manuscript should certainly involve additional markers; however, we used all the data available from the spadefoot toads sampled outside the South Texas population. Those data come from Rice & Pfennig (2008), and are based on only a single genetic marker. The DNA samples used in Rice & Pfennig (2008) are no longer available, meaning it is impossible for us to incorporate data from additional markers.

For these reasons, we hope that PeerJ might reconsider our manuscript. We have revised our manuscript to address all the other comments provided by the Reviewer and attached both a revised copy of the manuscript and our detailed response to the reviewer’s comments. It is not possible for us to incorporate additional genetic markers in our analysis, and we hope that a Handling Editor with expertise in species distribution modeling would recognize that the limited population genetic data we present are a useful and unique complement to the main focus of our paper.

Sincerely,

Amanda Chunco

Emma Nault

Rebecca Silverman

Sarah Midolo

Hanna Harper

Amber Rice


· · Academic Editor

Reject

I sent the manuscript out for review to determine whether the referee(s) would identify the merits of the study that would warrant publication for PeerJ. Although the reviewer recognized merits, s/he mentioned limitations and drawbacks, raising some misgivings about the way the manuscript has been analyzed. I agree with many of their statements and concerns, mainly regarding the use of only one uniparental genetic marker. Based on this point, I am sorry but I can unfortunately not recommend this paper for publication in PeerJ. I hope all advice can be helpful when reviewing your manuscript and I wish the authors good luck in seeing their work published elsewhere.

Reviewer 1 ·

Basic reporting

Overall, I found this to be a well written manuscript. I especially liked the analysis of multiple species to test whether the range disjunction is real or might reflect observational biases.

I think some of your language could be more direct. A few examples:

Here is your Abstract reworked a bit:

Range disjunctions appear to be common in nature, although they may be caused by various factors. They may simply be an artefact of inadequate sampling. If real, they may be the result of colonization events or inhospitable habitat. With natural habitats showing increasing fragmentation because of human activity, understanding the cause of a disjunction can also have important conservation implications. We investigate the geographical range of the Plains spadefoot toad, Spea bombifrons, a widely-distributed species in the midwestern United States, with a putative disjunct population in southern Texas. We combine GIS mapping, species distribution modeling, and population genetic analysis to investigate this putative disjunction. We establish that this southern Texas population is truly geographically disjunct and genetically distinct. Further, using climate projections we show that this unique population is at high risk of local extinction.

Line 31. ‘Range disjunctions occur when two or more independent populations …”

Lines 32-33. Instead of “This pattern of observed geographical isolation can occur for one of four reasons.” Perhaps “Geographical isolation can arise for four reasons:”

Line 37. Delete “only”

Line 38. I am an aficionado of AR Wallace and have first editions of many of his books, but I think you should simply say incomplete knowledge rather than Wallacean shortfall. Lines 52 and 53 as well. – change to gaps in our knowledge of distributions.

Line 41-42. Spell out what one overestimates with respect to the range – i.e. it is incorrectly assumed that the region between areas where a species is known to truly occur is occupied.

Line 52. Change ‘methodology ‘to ‘approach’

Lines 65 and 66. I think simply say that a range disjunction has been proposed to occur rather that saying herpetologists have proposed.

Line 67. Change to “Here we combine …” rather than “Here we use a combination of …”

Line 77. Delete ‘across the range’

Line 79. Dodd 2013 is one source. You mention sources. Perhaps provide one other at least.

Line 95. Data was but later Data were. Consistency as to whether you are using data as a collective noun.

Line 110. Change ‘could be indicative of’ to ‘could represent’

Line 111. Here and elsewhere – change anuran to all lower case. How about saying other frog and toad species instead to make it more accessible.

Line 136. I prefer to use the full genus name when beginning a sentence – thus Spea rather than S.

Line 156-157. Change ‘has been shown to improve’ to ‘improves’

Line 160-161. Change to ‘and then evaluated our models by comparing test and training AUC values (AUC = area under the receiver operating characteristic curve).’

Line 263. Presumable ‘regularization multipliers’ perhaps use ‘regularization multiplier values’

Line 347. I would delete the word ‘finally’

Line 350 (and elsewhere). Given that Gherghel & Martin (2020) apparently used niche modeling to explore glacial and postglacial range dynamics in spadefoot toads should this reference and findings therein not feature earlier? Perhaps in the Introduction?

Lines 353-354. ‘sweepstakes event’ is an unnecessary phrase. Just say a rare long-distance colonization event.

Lines 364-369. See my other comments on your population genetics analysis. I think you sample sizes and single marker approach limit what you can say on the genetics side. See too Lines 376-378.

Legend for Figure 1. I would suggest you be a bit more expansive in your description. Scenario A ‘where the current lack of distributional records reflects a real range disjunction’ and Scenario B ‘where the apparent gap reflects lack of adequate sampling and the range is contiguous.’

Figure 2. There seems to be a stray ‘a.’ in the left panel. For non-North America readers I would label Canada and the USA.

Table 3. I’d suggest too many significant digits for PhiST.

Experimental design

You have defined your research questions well.

As you will see in my comments on the validity of your findings, I think that mtDNA surveys alone are inadequate and you should include nuclear data/

Lines 213-225. Why not use all the sequences available? I think you need a more thorough explanation of your rationale here. At the least, for the haplotype network including all available data makes more sense to me. Why did you only randomly sample 17 individuals once? Would not repeated subsampling give more robust analyses? You delineate three ‘populations’ within the core range but would it not be preferable (assuming sufficient sampling intensity) to compare as many sampling locales/populations as possible and assess whether the putatively disjunct population exhibits proportionately greater genetic differentiation per unit pairwise geographic distance relative to all other core pairwise comparisons?

Validity of the findings

Lines 173-179. You have used one future climate scenario under RCP 4.5 assuming modest reductions in emissions. While your results do predict disappearance of the disjunct population in Texas, I do wonder whether exploring other scenarios might give you a more complete picture. How much time do we have under the status quo scenario with no change in emissions as seems more likely than reduction? Why choose 2070? Provide a rationale. Are there any scenarios under which this disjunct population would persist? Line 174. I do wonder whether you might rename the population genetic test data? It might cause some mild conflation of your two different approaches – niche modelling versus population genetics.

The mitochondrial sequence data are intriguing but insufficient for testing your hypotheses. There are lots of reasons that mtDNA alone should not be the sole source of genetic data to test for assertions regarding genetic distinctiveness: maternally-inherited, represent only a single marker, mito-nuclear discordance is not uncommon. I recognize that not everyone has the capacity to survey panels of single nucleotide polymorphisms but there are published microsatellites that could have been used. e.g. Rice et al. (2008), Development and characterization of nine polymorphic microsatellite markers for Mexican spadefoot toads (Spea multiplicata) with cross-amplification in Plains spadefoot toads (S. bombifrons). Molecular Ecology Resources, 8: 1386-1389. https://doi.org/10.1111/j.1755-0998.2008.02291.x

Figure 5. Relative to many other mtDNA surveys that I have seen of frog and toad populations I would say that the divergence among haplotypes here is quite modest. I think you might be putting too much stock in your summary Tajima’s D indices and the haplotype distributions based on relatively low sample numbers. Lines 304-315.

---

## Round 0.2 · Major Revisions

We have received two more sets of reviews for your manuscript. Both of them are positive, but especially reviewer 3 has provided a number of comments that need to be addressed.

Reviewer 2 ·

Basic reporting

Although I wasn't among the original reviewers, I find the authors' revisions, based on the feedback received from peer reviewers, quite compelling.

Experimental design

The authors' work was well taught and rigorously performed.

Validity of the findings

I am confused about their use of occurrence data for the Plains spadefoot toad up to 2018 only. I noticed that iNaturalist provides an excellent dataset for this species, and most of the records are after 2018. I wonder if there is any justification for limiting the available occurrence dataset to 2018. I also wonder whether not including additional records, for example, from undersampled areas of the range, might be influencing the results.

Reviewer 3 ·

Basic reporting

The manuscript contains an adequate amount of literature. However, it lacks a cohesive flow to effectively highlight the research gap. Additionally, the literature content present in the methods and results sections would be better suited for inclusion in the introduction. There are numerous grammar issues throughout the manuscript, including mixed tenses, unclear sentences, and sentences lacking a clear subject.
Need to share complete MaxEnt model results to see whether it is accurate or not.

Experimental design

the authors need to clearly articulate all aims of the project and demonstrate their interconnectedness. The research question is defined, relevant and meaningful. However, it needs more clarity. This research was done using some technical standards. The method is not clear for the MaxEnt species distribution model.

Validity of the findings

The authors mentioned the accuracy of the MaxEnt model. However, it is not sufficient. Need replications. There is no clear rationale. Need more results to prove the MaxEnt model is accurate, statistically sound and controlled.

Additional comments

Thank you for submitting your manuscript titled "Population Isolation in the Plains spadefoot toad: Causes and conservation implications." The authors present a valuable concept for assessing the spadefoot toad and exploring conservation measures to address current and future scenarios. However, while the overall idea of identifying the current and future distribution of the spadefoot toad is clear, the aims, methods, and results of this project lack clarity. I hope the following constructive comments will help you improve the quality of your project.
I highly recommend that the authors need to identify the objectives and aims of the project. It is not clear. This clarification is essential for better understanding.
I am confused about why the authors utilised two different techniques, species distribution modelling and population genetic analysis, to study the spadefoot toad. A strong rationale for their inclusion should be provided if both methods are necessary.
I suggest reading the MaxEnt model instruction manual before running the model will be helpful to you to get an understanding of the theory behind the MaxENT model. While reading the manuscript I found you have a less understanding of the species distribution model. It is not something you feed to software and get numbers as output and say it has high values for some parameters. As a solution, I recommend selecting single variables from highly correlated variables, such as Bio 5 (maximum temperature of the warmest month), Bio 8 (mean temperature of the wettest quarter), and Bio 10 (mean temperature of the warmest quarter). Running the model with replicates and replicate types would also be beneficial. Without this information, it is challenging to assess the quality of the model. Furthermore, conducting a jackknife test to evaluate the individual contribution of each variable to the model's prediction is advisable.
Based on the model prediction with a 0.5 regularization multiplier and the AUC value, as well as without jackknife test results, and response curves, it is difficult to determine whether these populations are indeed disconnected. Therefore, I strongly recommend improving the model and providing all relevant information as supplementary data.
In addition to issues with the modelling method, numerous academic English issues throughout the manuscript. The first two paragraphs of the method section should be moved to the introduction, as they contain more literature review than methodological steps. The section under population genetics needs to be rewritten in the proper verb form of a method section. It is unnecessary to include personal opinions or reflections; instead, focus on detailing what was done and how it was done. Several instances of mixed verb patterns throughout the manuscript should be thoroughly proofread and corrected.
I have provided examples of grammar issues in some places, but there are likely more throughout the manuscript.
109 environmentally disjunct?;
114 to 135 This is a part of the introduction, explaining S. bombifrons distribution, life cycle and genetic diversity. It is better to move to line 103.
121 will?
142-144 please rewrite. The latitudes and longitudes were assigned ………
147 Why the 10 km accuracy limit? What is the ecological/genetic factor to determine 10 km is the cut-off value?
171 Grammar
199-200 Grammar
207-208 Why did you use the Regularization multiplier from 0.2-5 range? How many models did you develop between these two values?

213 How many replicates did you use? What is the replicate run type?
221-223 Grammar
233-243 Most of the content must go to the introduction as literature. Please proofread carefully
239 Grammar
244-270 Please rewrite the section using the correct sentence pattern.
417 regularization multipliers. You have a regularization multiplier range from 0.2 to 5. What about the regularization multiplier 1 value map? What is the deviation from 1 to 0.5?
422-429 Not clear what is the lower value of the regularization multiplier.
430 Your model has a high AUC value because it has no replicates and has highly correlated abiotic variables. It doesn’t say your model is good. Please run a model with a reasonable regularization multiplier and at least 10 replicates with a proper replicate type.

---

## Round 0.3 · accepted · Accept

Thank you very much for addressing the reviewer comments in your revision. We have now received another set of reviews and based on their assessment of this revised manuscript I am happy to say that this manuscript is ready for publication now.

Reviewer 3 ·

Basic reporting

Excellent
Clear and unambiguous, professional English used throughout
Literature references, sufficient field background/context provided
Professional article structure, figures, tables. Raw data shared
Self-contained with relevant results to hypotheses

Experimental design

The research question is well-defined, relevant & meaningful. It is stated how research fills an identified knowledge gap. The aim and objectives are clear and easily understandable—methods are described with sufficient detail & information to replicate. Doing ten replicates with cross-validation made results more reliable.

Validity of the findings

Meaningful replication is encouraged where rationale & benefit to literature are clearly stated
underlying data have been provided; they are robust, statistically sound, & controlled
Conclusions are well stated, linked to the original research question & limited to supporting results

Additional comments

Thank you for addressing the reviewer's comments. The revisions have made the manuscript clearer.